# Gonadotroph Tumors Show Subtype Differences that Might Have Implications for Therapy

**DOI:** 10.3390/cancers12041012

**Published:** 2020-04-20

**Authors:** Mirela Diana Ilie, Alexandre Vasiljevic, Camille Louvet, Emmanuel Jouanneau, Gérald Raverot

**Affiliations:** 1Endocrinology Department, “Groupement Hospitalier Est” Hospices Civils de Lyon, 69677 Bron, France; mireladiana.ilie@gmail.com (M.D.I.); camille.louvet@ch-metropole-savoie.fr (C.L.); 2Endocrinology Department, “C.I.Parhon” National Institute of Endocrinology, 011863 Bucharest, Romania; 3Pathology Department, “Groupement Hospitalier Est” Hospices Civils de Lyon, 69677 Bron, France; alexandre.vasiljevic@chu-lyon.fr; 4Endocrinology Department, Centre Hospitalier Métropole Savoie, 73000 Chambéry, France; 5Neurosurgery Department, “Groupement Hospitalier Est” Hospices Civils de Lyon, 69677 Bron, France; emmanuel.jouanneau@chu-lyon.fr

**Keywords:** gonadotroph tumors, gonadotroph adenomas, tumor subtype, sex differences, tumor heterogeneity, estrogen receptor alpha (ERα), somatostatin receptor 2 (SST2), somatostatin receptor 5 (SST5)

## Abstract

Gonadotroph tumors, although frequent, are poorly studied and understood, being usually included in the larger nonfunctioning pituitary neuroendocrine tumors (PitNETs) group. Moreover, in comparison to the other types of PitNETs, no established medical treatment is currently available for gonadotroph tumors. Here, we performed a retrospective study and analyzed the clinicopathological characteristics of 98 gonadotroph tumors operated in a single large pituitary center. Although being larger in men (*p* = 0.01), the aggressiveness of gonadotroph tumors did not appear to be sex-related. LH tumors were rare (4/98) and exclusively encountered in men. Somatostatin receptor type 5 (SST5) was absent in all analyzed tumors. The immunoreactive score (IRS) of somatostatin receptor type 2 (SST2) and of estrogen receptor alpha (ERα) was associated with the histological subtype (*p* = 0.01 and *p* = 0.02). IRS ERα correlated moderately with IRS SST2 in all (rho = 0.44, adjusted *p*-value = 0.0001) and in male (rho = 0.51, adjusted *p*-value = 0.0002) patients, and with follicle-stimulating hormone (FSH) percentage in all (rho = 0.40, adjusted *p*-value = 0.0005) and in female (rho = 0.58, adjusted *p*-value = 0.004) patients. In conclusion, gonadotroph tumors exhibit histological characteristics pinpointing the existence of several subtypes. Their heterogeneity warrants further investigations and may have to be taken into account when studying these tumors and investigating treatment options.

## 1. Introduction

Pituitary adenomas, recently renamed pituitary neuroendocrine tumors (PitNETs) [1,2], are common neoplasms, representing 10–20% of all intracranial tumors. Of these, gonadotroph tumors represent approximatively one-third [3,4]. Although frequent, these tumors are poorly studied and understood, being usually included in the larger nonfunctioning (NF)-PitNETs group. Even though these tumors are usually clinically nonfunctioning and therefore not responsible for a hypersecretion syndrome, they are nonetheless frequently responsible for local compression and may be, depending on the degree of invasion of the surrounding structures, not completely removable by surgery. Moreover, following surgery, these tumors may regrow even when no tumor residue is visible [5]. 

Other than surgery, which remains the main treatment option, radiotherapy may sometimes be proposed, but in comparison to the other types of PitNETs, no established medical treatment is available or recommended for NF-PitNETs, including gonadotroph tumors [6]. As some of these tumors express dopamine receptor type 2 (D2R) [4,6] and somatostatin receptors (SSTs) [4], a role for D2R agonists and SSTs ligands has been envisioned. In the few studies published, these treatments showed some effect, but additional results from larger and long-term prospective randomized trials are necessary before recommending these medical options and, also, in order to establish whether there are subgroups of patients that would be better responders [4,6].

Regarding their aggressiveness, gonadotroph tumors appear to be less aggressive than other PitNETs. In the recent European Society of Endocrinology (ESE) survey on aggressive pituitary tumors and carcinomas, only 5 out of 125 aggressive PitNETs (4%) and only 1 out of 40 carcinomas (2.5%) were gonadotroph tumors [7]. However, for the cases that are aggressive, there is a lack of medical treatment options in comparison to other aggressive PitNETs: first, NF-PitNETs are less likely to respond to temozolomide than clinically functioning ones—only 17% of them showed regression on first-line temozolomide compared to 45% of clinically functioning PitNETs [7]; and, second, in comparison to other PitNET types, no case of a gonadotroph tumor treated with vascular endothelial growth factor receptor-targeted therapy, tyrosine kinase inhibitors, mammalian target of rapamycin (mTOR) inhibitors or immune checkpoint inhibitors, is reported in literature [8]. 

As for potential prognostic factors, one study reported that in male patients only, a negative (0 or 1) immunoreactive score (IRS) of estrogen receptor alpha (ERα) was associated with a higher reintervention rate and with an earlier reintervention [9]. These results, although not yet confirmed by others, highlight the need for a subclassification of gonadotroph tumors in order to better understand their heterogeneity and to improve their clinical management. This will imply, not only the study of the tumor cells themselves, but also the study of the tumor microenvironment [10].

Here, we aimed to phenotype gonadotroph tumors by studying the clinical (age, sex, dimension, invasion, complete surgical removal status, relapse, or progression) and pathological (expression of follicle-stimulating hormone (FSH) and/or luteinizing hormone (LH), ERα, SST2, SST5, proliferative markers) characteristics of 98 gonadotroph tumors operated in our expert pituitary center. We concluded that gonadotroph tumors exhibit histological characteristics pinpointing the existence of several subtypes and that this heterogeneity may have to be taken into account when studying these tumors, their origin, tumor microenvironment, behavior, prognosis and their response to treatment.

## 2. Results

### 2.1. Larger Gonadotroph Tumors in Men Compared to Women

Our study included 98 gonadotroph tumors, 64 from male patients and 34 from female patients, the sex ratio (M/F) being 1.9:1. The clinicopathological characteristics of these tumors, considered together and by patient gender are presented in Table 1. The immunohistochemical (IHC) expression of SST2 and ERα was assessed by our pathologist using the IRS, as illustrated in Figure 1.

The most frequent were the FSH-LH tumors, followed by the FSH tumors. LH tumors were rare (4/98) and exclusively encountered in men. SST5 was not expressed by any of the 21 analyzed tumors. Gonadotroph tumors were larger in male compared to female patients (*p* = 0.01), but their aggressiveness did not appear to be sex-related (there were no statistically significant differences regarding aggressiveness-related parameters, i.e., proliferation, invasion, grade, relapse or progression, between male and female patients).

### 2.2. Higher IRS SST2 in FSH Compared to FSH-LH and LH Tumors

When both male and female cases were considered together, FSH tumors demonstrated higher IRS SST2 than both FSH-LH and LH tumors (*p* < 0.05) (Figure 2A). When only male cases were analyzed, the trend was maintained, with a higher IRS SST2 in FSH tumors than both FSH-LH and LH tumors, but without reaching statistical significance (Figure 2B). FSH tumors also demonstrated higher IRS SST2 than FSH-LH tumors when only female cases were analyzed (*p* = 0.01) (Figure 2C).

Regarding the other clinicopathological characteristics tested, there were no statistically significant differences. Therefore, IRS SST2 associated with the histological subtype in all and in female cases, but not with the sex of the patients, nor with criteria of aggressiveness.

### 2.3. Higher IRS ERα in FSH Compared to LH Tumors

When both male and female cases were considered together, FSH tumors demonstrated higher IRS ERα than LH tumors (*p* < 0.05) (Figure 3A). When only male cases were analyzed, the trend was maintained, with a higher IRS ERα in FSH tumors than in LH tumors, but without reaching statistical significance (Figure 3B). There was no statistically significant difference for female cases, but at the same time, no LH tumors were present in this group (Figure 3C).

Regarding the other clinicopathological characteristics tested, there were no statistically significant differences. Therefore, IRS ERα associated with the histological subtype, but not with the sex of the patients, nor with criteria of aggressiveness. 

### 2.4. IRS ERα Correlates with IRS SST2 Especially in Men and with FSH Percentage Especially in Women

The results of the correlation analysis are presented in Table 2. There was a moderate correlation between IRS ERα and IRS SST2 in all patients (rho = 0.44, adjusted *p*-value < 0.0001), correlation that was stronger (rho = 0.51, adjusted *p*-value = 0.0002) when only male patients were considered, while it was only weak (rho = 0.32) and lost its statistical significance when only female patients were considered.

There was also a moderate correlation between IRS ERα and the percentage of FSH in all patients (rho = 0.40, adjusted *p*-value = 0.0005), correlation that was stronger (rho = 0.58, adjusted *p*-value = 0.004) when only female patients were considered, while it was only weak (rho = 0.31) and lost its statistical significance when only male patients were considered.

Therefore, IRS ERα correlated moderately with IRS SST2 in all and in male patients, and with FSH percentage in all and in female patients.

## 3. Discussion

Here, we performed a retrospective analysis of the clinicopathological characteristics of 98 gonadotroph tumors operated in our center. Gonadotroph tumors appear to have a male predominance [9,11,12], which we also found in our study. Regarding the histological subtype, we divided gonadotroph tumors in FSH-LH, FSH, and LH tumors. In the literature, other studies have also tried to subdivide gonadotroph tumors using similar approaches (with some variations, according also to the period in which they were performed) and found several subtypes [11,12,13,14]. However, to the best of our knowledge, this study is the first to show that these different subtypes show not only sex-related differences, but also that they associate with SST2 and ERα expression, which may be of biological and therapeutic importance.

The data available on the use of SSTs ligands in gonadotroph tumors is limited [4,15]. So far, studies including usually a small number of patients and various combinations of NF-PitNETs, more or less well characterized, showed that SSTs ligands octreotide and lanreotide had limited effect on tumor shrinkage [4,15,16,17,18]. In one of the studies showing no case of tumor shrinkage on octreotide LAR, the IHC expression of SST2 was investigated—as it was positive in only 46% of cases, the authors concluded that this may have contributed to the limited effect of octreotide [16]. 

In our cohort of gonadotroph tumors, we investigated IRS SST2 in 97 out of 98 patients. IRS SST2 ranged from 0 to 12 in both males and females, but it was negative (0 or 1) in the majority of cases. This is concordant with two other recent studies that also analyzed IRS SST2 using monoclonal antibodies in large cohorts of gonadotroph tumors (more than 100 tumors each), and found a negative IRS of 0 or 1 in the majority of cases and an IRS ≥4 only in a small number of patients [9,19]. In comparison to these studies, we also analyzed the distribution of IRS SST2 between different histological subtypes of gonadotroph tumors and our results show that SST2 is expressed differently by different histological subtypes of gonadotroph tumors. These results raise the question of whether this may also be the case for other SSTs and whether the histological subtype may be a predictor of response to SSTs ligands. Moreover, from a biological point of view, it suggests that different tumorigenesis mechanisms may be responsible for SST2+ versus SST2− gonadotroph tumors, as recently proposed for SST5+ versus SST5− corticotroph PitNETs [20,21].

The expression of SST5 was investigated in our study in 11 males and 10 females, in all of which it was absent. The two large studies previously mentioned also analyzed IRS SST5 and found an IRS SST5 either negative in all cases [9] or positive in only 2 cases [19]. Therefore, it appears that SST5 is only rarely expressed by gonadotroph tumors.

The second medical treatment option, D2R agonists, demonstrated so far to be more effective than SSTs ligands in controlling tumor growth (including tumor shrinkage) in a couple of studies that were mostly retrospective and included various combinations of NF-PitNETs [4,6,18,22,23,24,25,26]. As predictive factors of response to D2R agonists, so far the most investigated was the expression of D2R, that yielded inconsistent results [22,25,26]. Interestingly, in Batista et al. study, although ERα IHC expression did not associate with cabergoline responsiveness in a statistically significant manner, 53% of cases that showed tumor shrinkage on cabergoline were ERα+, while only 37% of cases that showed tumor stabilization on cabergoline and none of the cases that showed tumor growth on cabergoline were ERα+ [25]. A second study that investigated the IHC expression of ERα in 53 cases of NF-PitNETs (excluding silent somatotroph and silent corticotroph tumors) treated with D2R agonists, did not find a statistically significant association between ERα expression and D2R agonists responsiveness either [22]. However, IRS ERα was higher (mean = 2.4, SD = 2.6) in the “absence of tumor growth” group compared to “tumor growth” group (mean = 1.5, SD = 2), and at the same time, in the “tumor growth” group, no tumor out of 15 presented a % of ERα+ cells of ≥10%, in comparison to 3 out of 19 tumors in the “no change” group and 5 out of 19 tumors in the “shrinkage” group. Therefore, even though these 2 studies reported no significant association between ERα expression and D2R agonists responsiveness, it is too early to discard the expression of ERα as a predictor of D2R agonists responsiveness in gonadotroph tumors. More studies are needed, especially that in prolactinomas it was previously found that all D2R agonist-resistant tumors had low ERα expression [27].

In our cohort of gonadotroph tumors, we investigated IRS ERα in 98 patients. A positive IRS ERα of ≥2 was more frequently found (in 62.2% of cases) than in another large cohort of gonadotroph tumors (42%) [9]. However, Øystese et al. used in their study tissue microarrays and included also gonadotroph tumors that were negative for both FSH and LH, which may partly explain the differences, since another study that analyzed 21 tumors positive for FSH/LH found that 67% of them expressed ERα [28]. In the study of Øystese et al., the authors also found that in male gonadotroph tumor patients only, a negative IRS of ERα (0 or 1) was associated with a higher reintervention rate and with an earlier reintervention and that the absence of ER*α* together with a younger age predicted the need for reintervention in male patients [9]. In our study, IRS ERα did not associate with criteria of aggressiveness, including relapse or progression, neither in male, nor in female patients. However, in our study a follow-up longer than one year (median = 43 months) was available in only 42 males and 20 females, of which only 8 males and 3 females demonstrated relapse or progression, in comparison to their study where the follow-up was longer (median = 148.6 months) and where 37 out of 122 patients ended up by needing a reoperation and 10 patients radiotherapy [9]. 

In comparison to the existing studies, we also analyzed the distribution of IRS ERα between different histological subtypes of gonadotroph tumors, and our results show that ERα is expressed differently by different histological subtypes of gonadotroph tumors. Moreover, we found a moderate correlation between IRS ERα and FSH percentage in all and in female patients. There is not yet enough data to speculate whether these findings are of biological importance (for example whether different tumorigenesis mechanisms are responsible for these apparently different subgroups or whether they would behave or respond to treatment differently), but nonetheless further studies investigating this matter seem justified. 

Finally, we found a moderate correlation between IRS ERα and IRS SST2 in all and in male patients, while in female patients, the correlation was weak and did not reach statistical significance. This observation is in concordance with the study of Øystese et al. (sex ratio for gonadotroph tumors M/F = 1.9:1), which also found a moderate correlation between Er*α* and SST2 in both RT-qPCR and IHC analyses in male patients with gonadotroph tumors, but not in female patients [9]. Another study on 59 NF-PitNETs (excluding silent somatotroph and gonadotroph tumors) found the mRNA level of ER*α* to be correlated to that of SST2, as well [29]. Moreover, in breast cancer, the same correlation between the two receptors was found and, in addition, estrogen was shown to upregulate SST2 expression [30,31,32]. We speculate that the subgroup of patients having both a higher IRS SST2 and a higher IRS ERα may prove to respond differently to medical treatments (eventually combined) compared to the subgroup having both a lower IRS SST2 and a lower IRS ERα. In the future, treatment options for this group may potentially include combinations between ER-targeted therapy, including selective estrogen receptor modulators (SERMs), SSTs ligands and/or D2R agonists, like for example combinations of SERMs + SSTs ligands as recently tested in acromegaly [33] or chimeric molecules like TBR-760, a dopamine-somatostatin chimeric molecule, which is expected to enter soon into a Phase 2 clinical trial in patients with NF-PitNETs [34].

However, it will be of tremendous importance to not only stop considering gonadotroph tumors as simply NF-PitNETs (instead of a distinct group of tumors) and hormone-negative tumors as a homogenous group [2,3], but also to start acknowledging that gonadotroph tumors themselves are heterogeneous. Indeed, our study comes to add to the growing literature on the heterogeneity of PitNETs and on the need to continue to refine the histological classification of these tumors [3,21]. Very recently, Neou et al. used complex multi-omics analysis to demonstrate it. Our study, although having several limitations (its retrospective design, the small sample sizes in certain subgroup analysis—especially the presence of only 4 LH tumors, the lack of IHC testing of SST1, 3 and 4, as well as the fact that no technique other than histological/IHC analyzes were performed), it shows that even with readily available and inexpensive methods (such as IHC), these tumors can be subclassified, and we think that in the future as many authors as possible should strive to do the same.

## 4. Materials and Methods 

### 4.1. Gonadotroph Tumor Samples

Tumor tissue from 98 patients with gonadotroph tumors was obtained during surgery, which was performed in a single department: The Neurosurgery Department of *Groupement Hospitalier Est, Hospices Civils de Lyon*, Lyon, France, from June 2012 to March 2019. All tumors were diagnosed and classified by routine histopathological and IHC studies by the same pathologist, Dr. Alexandre Vasiljevic, in the Pathology Department of *Groupement Hospitalier Est, Hospices Civils de Lyon*, Lyon, France. The histological subclassification in FSH, FSH-LH and LH tumors was based on the IHC staining. The study was approved by the ethics committee of *Hospices Civils de Lyon*. Consent was obtained from each patient or subject after full explanation of the purpose and nature of all procedures used. Patient information was recorded in a local database (PITUICARE-Lyon, registered with the French data protection agency CNIL, 16-021, and clinicaltrials.org, NCT 02854228).

### 4.2. Clinicopathological Data

Clinical data (age at surgery time, sex, maximal diameter, complete surgical removal status, relapse or progression) and pathological data (proliferative markers, tumor subtype, FSH percentage, LH percentage, IRS ERα, IRS SST2, and IRS SST5) was retrospectively collected from the patient’s medical files. Invasiveness was classified according to the preoperative magnetic resonance imaging. In order to do so, the last magnetic resonance imaging exam before surgery was reviewed. A tumor was considered proliferative on the presence of at least 2 of the 3 criteria: Ki67 ≥ 3%, mitoses *n* > 2/10 high power fields (400× magnification), and positive p53 [35]. Tumors were then graded as following: grade 1a (non-invasive and non-proliferative tumor), grade 1b (non-invasive, but proliferative tumor), grade 2a (invasive, but not proliferative tumor), and grade 2b (invasive and proliferative tumor). Aggressiveness-related parameters were considered the proliferation, invasion, grade, and the relapse or progression.

### 4.3. Automated DAB Chromogen IHC

Formalin-fixed paraffin embedded tumors were cut in serial sections (4 µm). Immunohistochemical analysis was performed using a routine protocol on a BenchMark^®^ ULTRA automated immunostainer (Ventana Medical Systems Inc., Roche Diagnostics, Tucson, AZ, USA) in the Pathology Department of *Groupement Hospitalier Est, Hospices Civils de Lyon*, Lyon, France. The primary antibodies used were against β-FSH (mouse monoclonal, 1:3000, #0373, Beckman Coulter Immunotech, Marseille, France), β-LH (rabbit polyclonal, 1:8000, #AF-559518-89, gift from the National Institute of Diabetes and Digestive and Kidney Diseases, Bethesda, MD, United States), Ki67 (mouse monoclonal, clone MIB-1, 1:100, #M7240, Agilent Dako, Santa Clara, CA, United States), SST2 (rabbit monoclonal, clone UMB-1, 1:4000, #ab134152, Abcam, Cambridge, UK), SST5 (rabbit monoclonal, clone UMB-4, 1:500, #ab109495, Abcam, Cambridge, UK) and ERα (rabbit monoclonal, clone SP1, prediluted, Ventana Medical Systems Inc., Roche Diagnostics, Tucson, AZ, USA). Bound antibodies were detected using a Ventana kit including DAB reagent (ultraView Universal DAB Detection Kit, Ventana Medical Systems Inc., Roche Diagnostics, Tucson, AZ, USA). Slides were next counterstained in hematoxylin and finally bluing was performed using the Bluing Reagent (Ventana Medical Systems Inc., Roche Diagnostics, Tucson, AZ, USA). 

### 4.4. IRS ERα, SST2 and SST5

IRS was calculated as the percentage of positive nuclei (0 to 4: 0 if 0%, 1 if ≤10%, 2 if 11–50%, 3 if 51–79% and 4 if ≥80%) × the intensity of the staining (0 to 3: 0 if no staining, 1 if mild, 2 if moderate and 3 if strong), as previously described [27].

### 4.5. Statistical Analysis

Statistical analysis was carried out using R commander of R package version 3.6.1 (R-project, Vienna, Austria) and GraphPad Prism 5 (GraphPad, San Diego, CA, USA). Student’s *t*-test, Mann–Whitney test, Kruskal–Wallis test followed by Dunn’s post-test, chi-squared, Fischer’s exact test, and Spearman’s correlation were applied as appropriate. *p* < 0.05 was considered significant (in the case of the correlation matrix, adjusted *p*-value was calculated using the Holm’s method). Graphs were made using GraphPad Prism 5 (GraphPad, San Diego, CA, USA).

## 5. Conclusions

Although being larger in men, in our study the aggressiveness of gonadotroph tumors did not appear to be sex-related. IRS ERα and IRS SST2 were associated with the histological subtype and IRS ERα correlated moderately with IRS SST2 in all and in male patients. LH tumors were rare, exclusively encountered in men and negative for both ERα and SST2. Therefore, we showed that gonadotroph tumors exhibit histological characteristics pinpointing the existence of several subtypes, which may be of biological and therapeutic importance. 

Our study comes to add to the growing literature on the heterogeneity of PitNETs and on the need to continue to refine the histological classification of these tumors. In order to better understand gonadotroph tumors and to improve their clinical management, this will imply not only to stop considering gonadotroph tumors as simply NF-PitNETs (instead of a distinct group of tumors) and hormone-negative tumors as a homogenous group, but also to start acknowledging that gonadotroph tumors themselves are heterogeneous. This is of tremendous importance because different subtypes of gonadotroph tumors may prove to have different tumorigenesis mechanisms, different composition of their associated tumor microenvironment or different behavior. Furthermore, these differences may translate into different prognosis and response to treatment. Therefore, the heterogeneity of gonadotroph tumors warrants further investigations and may have to be taken into account when studying these tumors, their origin, associated tumor microenvironment, behavior, prognosis and their response to treatment.

## Figures and Tables

**Figure 1 cancers-12-01012-f001:**
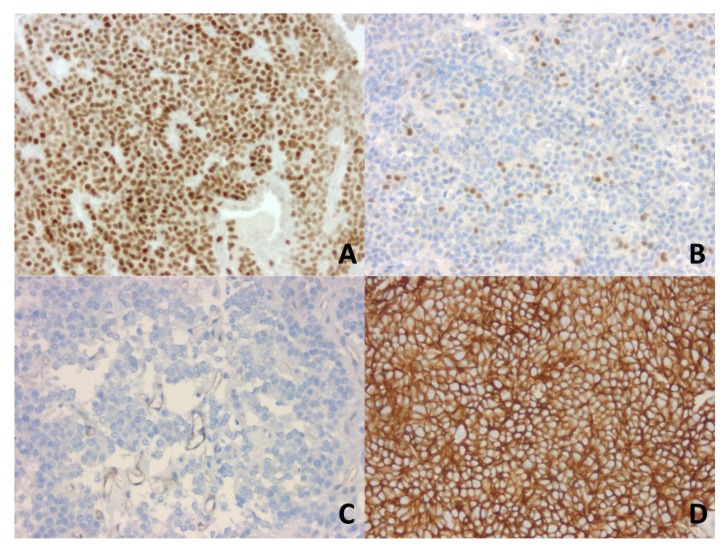
Immunohistochemical analysis of estrogen receptor alpha (ERα) and of somatostatin receptor 2 (SST2) expression in gonadotroph tumors using the immunoreactive score (IRS). IRS is calculated as the percentage of positive nuclei (0 to 4) × the intensity of the staining (0 to 3) (original magnification, ×200). (**A**). IRS ERα = 12 in a FSH-LH tumor (strong nuclear immunoexpression in 95% of neoplastic cells). (**B**). IRS ERα = 2 in a FSH-LH tumor (moderately intense immunopositivity in 5% of neoplastic cells). (**C**). IRS SST2 is negative in this FSH-LH tumor (absence of immunopositivity in the neoplastic cells; faint vascular expression as a positive internal control). (**D**). IRS SST2 = 12 in a FSH tumor (strong membranous immunoexpression in 100% of neoplastic cells). Abbreviations: follicle-stimulating hormone (FSH), luteinizing hormone (LH).

**Figure 2 cancers-12-01012-f002:**
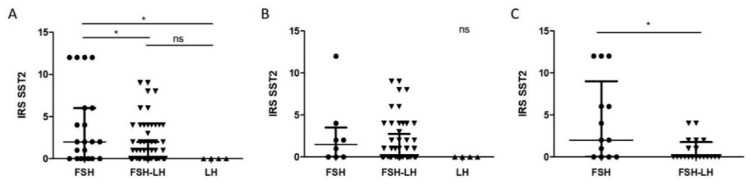
Tumor subtype-related comparison of IRS SST2 distribution in all (**A**), male (**B**) and female (**C**) cases. Graphs: median with interquartile range; each individual point represents a tumor: circles represent FSH tumors, down-pointing triangles FSH-LH tumors, and up-pointing triangles LH tumors. (**A**). Statistical test: Kruskal–Wallis, *p* = 0.01, *n* = 21, 72, 4, with Dunn’s test for columns comparison. (**B**). Statistical test: Kruskal–Wallis, *p* = 0.15, *n* = 8, 52, 4. (**C**). Statistical test: Mann–Whitney, *p* = 0.01, *n* = 13, 20. Abbreviations: immunoreactive score (IRS), somatostatin receptor 2 (SST2), follicle-stimulating hormone (FSH), luteinizing hormone (LH), not significant (ns), *p* < 0.05 (*).

**Figure 3 cancers-12-01012-f003:**
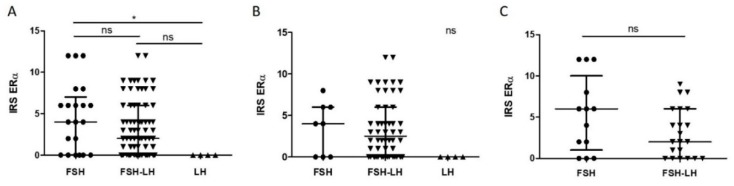
Tumor subtype-related comparison of IRS ERα distribution in all (**A**), male (**B**) and female (**C**) cases. Graphs: median with interquartile range; each individual point represents a tumor: circles represent FSH tumors, down-pointing triangles FSH-LH tumors, and up-pointing triangles LH tumors. (**A**). Statistical test: Kruskal–Wallis, *p* = 0.02, *n* = 21, 73, 4, with Dunn’s test for columns comparison. (**B**). Statistical test: Kruskal–Wallis, *p* = 0.057, *n* = 8, 52, 4. (**C**). Statistical test: Mann–Whitney, *p* = 0.19, *n* = 13, 21. Abbreviations: immunoreactive score (IRS), estrogen receptor alpha (ERα), follicle-stimulating hormone (FSH), luteinizing hormone (LH), not significant (ns), *p* < 0.05 (*).

**Table 1 cancers-12-01012-t001:** Sex-related comparison of clinicopathological characteristics of the 98 gonadotroph tumors included in the study.

Clinicopathological Characteristics	All Tumors	Males	Females	*p*-Value (Males vs. Females)
Age at surgery time (years) (*n* = 98/64/34)	63 (33–86)	63 (37–86)	61 (33–80)	0.44
Maximal diameter (mm) (*n* = 98/64/34)	27 (13–51)	28 (13–51)	24 (13–41)	**0.01**
Histological subtype (*n* = 98/64/34)				
FSH	21 (21.4%)	8 (12.5%)	13 (38.2%)	ND
FSH-LH	73 (74.5%)	52 (81.25%)	21 (61.8%)	
LH	4 (4.1%)	4 (6.25%)	0 (0%)	
Ki67 index (*n* = 98/64/34)				
<3	84 (85.7%)	55 (85.9%)	29 (85.3%)	0.93
≥3	14 (14.3%)	9 (14.1%)	5 (14.7%)	
Mitoses (*n* = 98/64/34)				
≤2	89 (90.8%)	60 (93.75%)	29 (85.3%)	0.26
>2	9 (9.2%)	4 (6.25%)	5 (14.7%)	
p53 (*n* = 98/64/34)				
negative	60 (61.2%)	40 (62.5%)	20 (58.8%)	0.72
positive	38 (38.8%)	24 (37.5%)	14 (41.2%)	
Proliferation (*n* = 98/64/34)				
No	84 (85.7%)	56 (87.5%)	28 (82.4%)	0.48
Yes	14 (14.3%)	8 (12.5%)	6 (17.6%)	
Invasion (*n* = 94/62/32)				
Yes	53 (56.4%)	37 (59.7%)	16 (50%)	0.37
No	41 (43.6%)	25 (40.3%)	16 (50%)	
Grade (*n* = 94/62/32)				
1a	35 (37.2%)	23 (37.1%)	12 (37.5%)	ND
1b	6 (6.4%)	2 (3.2%)	4 (12.5%)	
2a	47 (50%)	32 (51.6%)	15 (46.9%)	
2b	6 (6.4%)	5 (8.1%)	1 (3.1%)	
IRS ERα (*n* = 98/64/34)	2.5 (0–12)	2 (0–12)	3.5 (0–12)	0.35
IRS SST2 (*n* = 97/64/33)	0 (0–12)	0 (0–12)	0 (0–12)	0.58
IRS SST5 (*n* = 21/11/10)	0 (0–0)	0 (0–0)	0 (0–0)	ND
Complete surgical removal (*n* = 93/60/33)				
Yes	57 (61.3%)	36 (60%)	21 (63.6%)	0.73
No	36 (38.7%)	24 (40%)	12 (36.4%)	
Follow-up > 1 year (months) (*n* = 62/42/20, of which *n* = 2/1/1 had adjuvant RT)	43.0 (12.6–88.2)	46.3 (12.6–88.2)	37.6 (14.1–69.2)	
Relapse or progression > 1 year (n = 60/41/19)				
Yes	11 (18.3%)	8 (19.5%)	3 (15.8%)	1
No	49 (81.7%)	33 (80.5%)	16 (84.2%)	

Age at surgery time, maximal diameter, IRS ERα, IRS SST2, IRS SST5, and the follow-up > 1 year are expressed as median with range (applied statistical tests: independent samples *t*-test for the age at surgery time and Mann–Whitney for the maximal diameter, IRS ERα and IRS SST2; for the rest of the parameters, either Chi-square or Fisher’s exact test were applied, as appropriate). Invasion refers to the radiological invasion. Abbreviations: follicle-stimulating hormone (FSH), luteinizing hormone (LH), number of patients for which the data is available (*n*), immunoreactive score (IRS), estrogen receptor alpha (ERα), somatostatin receptor (SST), radiotherapy (RT), non-determinable (ND).

**Table 2 cancers-12-01012-t002:** Correlation matrix between IRS ERα, IRS SST2, FSH (%), LH (%), age and diameter in all, male and female cases.

Clinicopathological Characteristics	IRS SST2	FSH (%)	LH (%)	Age (years)	Diameter (mm)
	Spearman’s Rho/Adjusted *p*-Value (Holm’s Method)/*n*				
IRS ERα					
All	0.44/<0.0001/97	0.40/0.0005/98	−0.05/1/98	0.06/1/98	0.04/1/98
Males	0.51/0.0002/64	0.31/0.16/64	0.01/1/64	0.11/1/64	0.04/1/64
Females	0.32/0.83/33	0.58/0.004/34	−0.16/1/34	−0.006/1/34	0.13/1/34
IRS SST2					
All		0.23/0.24/97	−0.13/1/97	0.23/0.26/98	−0.19/0.65/98
Males		0.18/1/64	−0.02/1/64	0.29/0.24/64	−0.28/0.29/64
Females		0.32/0.82/33	−0.36/0.55/33	0.10/1/33	0.03/1/33
FSH (%)					
All			−0.04/1/98	0.13/1/98	0.02/1/98
Males			0.10/1/64	0.18/1/64	0.04/1/64
Females			−0.30/0.92/34	0.05/1/34	0.07/1/34
LH (%)					
All				0.17/0.90/98	0.08/1/98
Males				0.13/1/64	−0.01/1/64
Females				0.18/1/34	0.02/1/34
Age (years)					
All					−0.03/1/98
Males					−0.12/1/64
Females					0.12/1/34

Abbreviations: immunoreactive score (IRS), estrogen receptor alpha (ERα), somatostatin receptor 2 (SST2), follicle-stimulating hormone (FSH), luteinizing hormone (LH), number of patients for which the data is available (*n*).

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
