# Peer review of "Gonadotroph Tumors Show Subtype Differences that Might Have Implications for Therapy"

_cancers, 2020, doi:10.3390/cancers12041012_

Round 1
Reviewer 1 Report
Thank you for the opportunity to read and review this manuscript.
The results are interesting and could point to a medical therapy in in selected patients with residual tumor after pituitary surgery.
I have some comments
Consider changing the title of the manuscript, what about?
“Gonadotoph tumors show subtype differences that might have implications for therapy”, since the main point here may not be related to gender?
Abstract line 25 an “was” is missing?
Line 25 “Molecules” is not wrong but consider rewriting
Table 1: all tumors are labeled SSTR2 negative. This must be a mistake?
Line 103 to 109 seems to be the table legend misplaced in the middle of the text? Line 102 seems to continue at line 109 with LH tumors, …..
The conclusion should start with the main findings described from line 304, and not with the rare LH tumors.
Reviewer 2 Report
The authors present a retrospective clinico-pathological analysis on a rare tumor entity providing evidence on the existence of heterogeneity and proposing several tumor subtypes. In my view, the manuscript is of interest to the clinical community and summarizes well valuable information; therefore I believe it would be of interest to the readership of Cancers. However, there are some minor points that need to be addressed. The manuscript would benefit from minor revision and clarification with respect to its limitations.
Minor comments:
Line 87: please specify aggressiveness-related parameters at this point, and also in methods section.
Line 103-106: please move applied statistical tests to methods section and table 1 caption instead. Accordingly, abbreviations in line 107-109 should be moved to table and figure 1 caption.
Line 201-204: long sentence and meaning difficult to follow.
Please address potential limitations of the study in discussion section, including its retrospective nature, small sample sizes in certain subgroup analyses, and lack of IHC testing of other SSR types.
Please elaborate on clinical and therapeutic implications of the study in the conclusions section.
Reviewer 3 Report
The authors present data on the immunohistochemical and clinicopathological characterisation of gonadotropin-producing pituitary adenomas.
The authors have only performed histological/immunohistochemical analyses on the archived samples, no other technique (e.g. molecular biological) has been implemented. The results show that these tumors show differences, but the clinical and biological relevance of these findings are difficult to interpret. Although the manuscript is well-written, in my opinion, there is no clear message to the reader.
It is not clear what implications for therapy do these findings deliver, as stated in the title.
Round 2
Reviewer 3 Report
The manuscript has improved.